# Sometimes Littering Is Acceptable—Understanding and Addressing Littering Perceptions in Natural Settings

**Naama Lev** [1,*]**, Maya Negev** [2] **and Ofira Ayalon** [1]

1 Department of Natural Resources and Environmental Management, Faculty of Social Sciences, University of Haifa, 199 Abba Hushi Blvd, Mount Carmel, Haifa 3498838, Israel; oayalon@univ.haifa.ac.il

2 School of Public Health, Faculty of Social Welfare and Health Sciences, University of Haifa, 199 Abba Hushi Blvd, Mount Carmel, Haifa 3498838, Israel; mnegev@univ.haifa.ac.il

\* Correspondence: levnaama@gmail.com

**Abstract:** The detrimental impact of visitor-induced litter pollution on ecosystems, wildlife, and overall quality of life emphasizes the urgency of mitigating it. This study uniquely focuses on diverse visitors' perceptions of littering behavior in open spaces, facilitating comprehensive assessment and targeted mitigation strategies. This study aimed to analyze attitudes, willingness to act, and responsibility perceptions, considering diverse demographics in Israel's multicultural context. It sought insights into littering rationales, potential remedies, and the identification of relatively acceptable littering behaviors for focused attention. This profound comprehension is crucial for conserving ecologically sensitive open areas, necessitating optimized management for interface preservation. Leveraging insights from an online survey involving 401 recent open-space visitors, this research reveals a disparity between self-professed and actual littering practices. Intriguingly, 32% of participants who claimed never to litter described instances of doing so. Furthermore, disparities emerged between anti-litter attitudes, willingness to act, and individual accountability, which were influenced by demographic variables. While individuals from various demographic cohorts attested to littering behavior, young ultra-Orthodox Jews possessing solely a high school level of education exhibited a proclivity for increased littering. Perceptions predominantly attribute purposeful and recreational motives to littering, rather than substantial reasons. Participants conceive a diverse range of effective strategies to address the issue, highlighting its intricate and multifaceted nature. Consequently, this study advocates for a multifaceted approach combining enhanced enforcement, educational campaigns, informative initiatives, and infrastructural enhancements. By acknowledging the complexities of littering behavior and embracing multifarious interventions, policymakers can enhance the likelihood of successfully curbing this pervasive challenge.

**Keywords:** littering behavior intention; attitudes toward littering; littering justification; open spaces; public policy; system justification theory

## 1. Introduction

Despite extensive efforts to curb littering, it remains persistent globally, necessitating comprehensive research for evidence-based solutions concerning littering and related environmental behaviors [1–4].

Littering in public spaces and natural environments is a widespread occurrence [3,5–8]. Increased human population, urbanization, and affluence have led to greater waste generation, particularly biodegradable waste, resulting in elevated litter pollution on streets, beaches, parks, and reserves [1,4,9–11]. Human behavior, predominantly individual littering, is a major contributor to this issue [3,12]. Despite extensive research studies, plans, and actions to stop litter behavior and prevent litter pollution, littering is still common in many places around the world [2,4,13]. Therefore, litter behavior and other environmental behaviors require further comprehensive research and understanding in order to develop evidence-based effective solutions.

The reduction in open and natural landscapes is a conspicuous trend, accentuating their inherent importance and appearing as one of the main sustainable development goals of United Nations [14]. The litter pollution of these areas with debris significantly impairs the ecosystems they host and subsequently, negatively infect public health and the overall quality of life [15]. In this vein, the current study focuses on littering behaviors within open spaces in urban cores and their outskirts.

Israel, a Western country situated in the Middle East, embodies a multifaceted, multicultural, and pluralistic society marked by coexisting yet occasionally conflicting tendencies, lifestyles, and behaviors [16,17]. This study engaged a representative sample of the Israeli population, facilitating an extensive exploration of each subgroup's perspectives on littering causation and effective mitigation strategies. Additionally, this inquiry sheds light on how these intricate social dynamics manifest in the characteristics of the littering phenomenon. Examining littering within Israel, given its unique attributes, promises to enhance our comprehension of littering and the interplay of social and individual determinants.

Research has revealed a disparity between anti-littering attitudes, a proclivity for cleanliness, and self-reported behaviors [18]. Despite recognizing the impropriety of littering, many individuals persist in this conduct, often resorting to justifications or rationalizations as coping mechanisms [19]. Unfortunately, this coping mechanism amplifies when littering occurs without rebuke, thereby exacerbating littering tendencies [20]. Identifying socially acceptable behaviors related to littering holds significant importance, especially among individuals who acknowledge their participation in littering, among others. Gaining insights into these behaviors can assist decision makers in developing appropriate strategies to effectively address the issue.

Littering is a behavioral problem that requires urgent solutions [12]. Studies have shown that the way to deal with the littering problem is to combine means by improving infrastructure, legislation and enforcement, advocacy, and education in the field [21,22]. As an illustration, it was determined that enforcement exhibited greater efficacy in clean environments compared to polluted ones [23]. However, the success of such means depends on their adaptation to and understanding of the local culture with respect to knowledge, beliefs, politics, morals, law, customs, and habits [8,24]. Environmental managers should engage applied social and environmental psychologists or social scientists to design such behavior-changing programs [12]. Therefore, it is extremely important that we deepen our understanding and examine littering behavior using new approaches and new locations.

## 1.1. Personal Factors Contributing to Littering

Littering is a negative environmental behavior that is very difficult to understand [25]. The intricacies of littering encompass a spectrum of influences, encompassing external aspects such as cleanliness levels, bin availability, and distribution [1,3,26], alongside personal determinants spanning sociodemographic and psychological dimensions [2,27–29]. Sociodemographic parameters, encompassing gender, age, religiosity, and education, have been correlated with littering tendencies. Notably, youth, males, individuals with greater religiosity, lower education, and reduced income exhibit a higher propensity for littering [1,3,11,30–36]. Psychological factors, such as awareness, attitudes and beliefs, willingness to act, self-responsibility, and locus of control and norms, have been examined in several studies on environmental behavior. Some suggest that attitudes are a significant factor affecting littering behavior in the public domain [30,37]. Stronger positions against littering and in favor of maintaining cleanliness were correlated with lower declared littering behavior. Moreover, awareness, willingness to act, and self-responsibility were negatively correlated to littering [2,27,38–40]. Examining environmental attitudes is important for assessing the impact of understanding environmental problems and attitudes toward personal littering.

## 1.2. Theoretical Framework—The Theory of Planned Behavior and Justification of Littering

Previous research has elucidated littering behavior through the theory of planned behavior, which posits behavioral intention as the primary precursor to behavior [24,41–44]. The most important prior factors that influence intention to perform a certain behavior are attitudes, norms, and locus of control. Over the years, models of human behavior have proven useful for understanding, predicting, and examining factors influencing human behavior [45]. Such models have been refined to include sub-factors of the predictive variables connected to littering [24,42,43,46]. An integrative model of justified behavior, combining and integrating processes postulated by both the neutralization theory and the theory of planned behavior [44]. This theory portrays the delinquent as an individual who subscribes generally to the morals of society but who is able to justify his own delinquent behavior through a process of "neutralization", whereby the behavior is redefined to make it morally acceptable [2]. It also includes attitudes, norms, knowledge, restrictions and options, habit formation, and evaluative processes of justification as determinants of behavioral decision making.

System justification theory posits that individuals when making immoral decisions for themselves instead of moral choices for the environment, necessitate justification [47,48]. In this context, a dilemma emerges, and these individuals encounter cognitive dissonance. Upon establishing justifications for their behavior, individuals are inclined to engage in it, as seen in behaviors like occasional littering. Justification mechanisms have also been explored as facilitating factors for environmentally detrimental behaviors, such as littering, thereby offering insights into the disparities between attitudes and actions [2]. Decision makers can learn about failures in environmental management following the justifications that arise. Through environmental education campaigns or activities, they can counteract cognitive justifications for littering to reduce it [2].

## 1.3. Littering and Dirt in Open Spaces and the Public Domain in Israel

Open spaces in Israel include green areas within the city, such as parks and urban nature sites, and open landscapes outside the city, including nature reserves, national parks, and beaches. These spaces are governed by different agencies and organizations: the Israel Lands Administration, the Jewish National Fund (JNF), the Israel Nature and Parks Authority (INPA), and municipal authorities [49]. Natural sites, particularly those outside cities, are considered to have higher sensitivity for littering because of the vulnerability of wildlife species that can immediately be harmed. In addition, the accessibility, cleaning facilities, and waste collection infrastructure for maintaining these areas are more complex than those for urban parks and nature [10].

Quantitative investigations pertaining to littering in open spaces are scarce, both in Israel and the broader context. Nevertheless, reports detailing the waste accumulation by Israelis at recreational sites are regularly featured in local media, particularly following national holidays. Furthermore, the diminishing emphasis on cleanliness in recent times has led to the persistence of unclean public spaces, spanning urban locales and natural environments [50]. Furthermore, despite the implementation of the cleanliness law in Israel since 1984 [51] and the provision for fines upon conviction for littering by individuals, organizations, municipalities, and others, the prominence of cleanliness in policy agendas remains deficient. Regrettably, it is often perceived as an outdated concern confined to aesthetics, underscoring a dearth of inter-organizational and inter-sectoral collaboration, suboptimal enforcement, inadequate financial resources, and a paucity of precise, quantitative insights into the magnitude of littering [50,52].

Society and culture significantly influence littering behavior; therefore, we must consider Israel's two major ethnonational groups—Israeli Jews and Israeli Arabs—representing about 79% and 21% of the population, respectively [17]. A significant minority group in Israeli society is the Jewish ultra-Orthodox sector, approximately 12% of the population, which profoundly differs from the majority group in terms of its religious, social, and cultural values, as well as ideologies and constructs, such as geographically different areas

and settings and using separate social and cultural networks (schools, educational organizations, and religious institutions) and mass media, social networks, and other channels [17,53]. Given these considerations, the principal research inquiry revolves around ascertaining the public's perceptions concerning littering within natural sites in Israel.

### 1.4. Objectives and Hypoteses

This study aims to achieve four objectives and corresponding research hypotheses:

1. Examine perceptions of littering encompassing anti-littering attitudes, perceived importance of cleanliness, environmental perspectives, willingness to act, self-responsibility, and self-perceived littering. Investigate correlations between declared littering and these factors.

**H1.** Participants prioritize anti-littering attitudes over environmental viewpoints.

**H2.** Participants exhibit stronger alignment with anti-littering attitudes compared to willingness to act and self-responsibility.

**H3.** Despite anti-littering attitudes, individuals still litter in the open spaces.

2. Examine demographic variations in declared littering behavior and attitudes toward littering within Israel.

**H4.** Attitude and behavior disparities exist across demographic facets (e.g., age, education, religious commitment). Younger, less educated, and more religious individuals tend to admit to littering with relatively moderate anti-littering attitudes.

3. Identify the primary cause of littering behavior and perceptions of strategies to mitigate it.

**H5.** Significant disparities in conceptions of littering rationales exist based on self-reported littering frequency. Frequent litterers attribute greater influence on external factors compared to intrinsic motivations.

**H6.** Public preference favors external interventions, like infrastructure maintenance and regulations enforcement, over transformative value and attitude shifts through educational initiatives.

4. Identify more "acceptable" and "normative" littering behaviors and categorize the general littering norm into sub-norms for deeper explanatory insight.

## 2. Methodology

To examine public perceptions, attitudes, willingness to act, and declared littering behavior, a survey was conducted in September 2020. The online questionnaire was distributed to ensure representation across the Israeli population. Ethical guidelines were strictly followed in accordance with the protocols of the Faculty of Social Science Ethics Committee (053/20).

### 2.1. Sample Design, Data Acquisition, and Construction of the Questionnaire

The study's target population comprised all individuals above 18 years of age in Israel who had visited a natural site in the past year. Only respondents with recent experience in nature sites were eligible to participate. Sampling was executed via two-stage systematic random sampling and a stratified-proportionate approach. This approach was chosen to encompass the diverse age, gender, and religious divisions characteristic of the Israeli populace. Filtering questions collected demographic data to ensure that the questionnaire participants formed a representative sample of the public.

The sample size was determined proportionately based on population density, assuming a normal distribution with a 5% sampling error, necessitating a minimum of 385 participants. Data collection was entrusted to the survey company "SekerNet," which manages an online access platform hosting more than 50,000 active panel members aged 18 and above, serving as representatives of the Israeli public. The company holds demographic data on its members and randomly invites participation while adhering to participation quotas aligned with demographic proportions derived from the Central Bureau of Statistics.

The questionnaire had 4 sections consisting of 60 quantitative closed questions and 2 qualitative open questions. The initial section inquired about littering frequency, where participants initially indicated their littering frequency in the past year. As individuals often claim to never litter, following this question, respondents were informed that littering occurs universally and occasionally. They were then asked to recall and honestly describe a personal instance of littering in nature sites, along with an incident when they observed others littering in such areas. Other questions related to the type of open space and habits, the last visit in nature, the importance of cleanliness, and litter types discarded by oneself and others, as well as reasons underpinning such actions. This section predominantly drew inspiration from an American questionnaire [3].

The second section examined the motivations behind littering and optimal strategies for addressing it, informed by Israeli and Jordanian investigations [24,38]. The third section comprised 42 statements assessing attitudes toward littering, environmental concerns, proactive inclinations, and personal accountability. These statements were adapted from prior studies, including modifications and additions to suit the context [7,35,37]. The final section gathered sociodemographic data from participants. The questionnaire is available upon request from the authors. The quantitative survey encompassed various variables (see Table 1).

**Table 1.** Dependent and independent research variables.

| Variable | Variable Type | Structure of the Question | Reliability (Cronbach's $\alpha$) | Example from Questionnaire |
|---|---|---|---|---|
| **Littering frequency** | Dependent | Based on 2 questions. 1. A quantitative question indicating frequency with 4 options (often, sometimes, seldom, never) 2. A qualitative question where participants were asked to honestly describe one time that they littered in nature | | *In recent years, how often do you litter (in streets, parking lots, parks, or nature)? We have all littered once in nature (the beach, city park or nature reserve), with a "hand on your heart", can you describe a specific case and explain why you did it?* |
| **Littering attitudes** | Independent | 10 statements; participants were asked to rate their consent on a 1–5 Likert scale (1 strongly disagree—5 strongly agree) | 0.863 | *I believe littering in nature is a negative habit. Litter harms the environment and nature.* |
| **Environmental attitudes** | Independent | 8 statements; participants were asked to rate their consent on a 1–5 Likert scale (1 strongly disagree—5 strongly agree) | 0.737 | *Mankind has the right to utilize natural resources according to its needs. Environmental issues have a direct impact on my daily life.* |
| **Willingness to act for cleanliness in nature** | Independent | 12 statements; participants were asked to rate their consent on a 1–5 Likert scale (1 strongly disagree—5 strongly agree) | 0.86 | *If I had enough time or money, I would definitely donate some of it to cleanliness in public space and nature. When the bin is full, I take the waste to another bin that has free space for disposal.* |
| **Self-responsibility** | Independent | 6 statements; participants were asked to rate their consent on a 1–5 Likert scale (1 strongly disagree—5 strongly agree) Participants were asked a question about who is responsible for keeping nature sites clean (the INPA, visitors, or both) | 0.521 | *Cleanliness in nature is the responsibility of the organization that manages the site. The presence of children near me makes me throw waste in the bin. Who do you think is responsible for cleaning nature sites and parks?* |

**Table 1.** *Cont.*

| Variable | Variable Type | Structure of the Question | Reliability (Cronbach's α) | Example from Questionnaire |
|---|---|---|---|---|
| **Importance of clean nature site** | Independent | Participants were asked to rate the importance of clean nature on a 1–5 Likert scale (1 not important at all—5 very important) | 0.86 | *How important is cleanliness to you at the nature site you visit?* |
| **Sociodemographic parameters** | Independent | Gender, age, religious affiliation, self-definition of the level of religiosity, education level, and income level | | |
| **Nature site type** | Independent | Participants were asked the name of the nature site they last visited. The sites were classified into five types: 1. Closed natural reserves with restricted entry hours and entrance fees; 2. Natural reserves open 24 h; 3. Planted forests and parks; 4. Lake and sea beaches; 5. Urban nature sites. | | |

### 2.2. Validity and Reliability of the Research Tool

The validity and reliability of the research tool developed for this study were examined in three stages. (1) Expert validation encompassed five specialists in statistics, litter behavior, and linguistics who reviewed the questionnaire for question details, style, wording, and completion time. Additionally, five individuals from unrelated fields reviewed the questionnaire to identify misunderstandings, complexities, and completion duration, resulting in necessary modifications. (2) In the subsequent phase, an initial questionnaire was disseminated through online social networks to a cohort comprising 62 participants. The inner consistency of variables was evaluated using Cronbach's alpha reliability, yielding satisfactory reliability values for all metrics, leading to adjustments that align with the language and terminology commonly employed in Israel. (3) Questionnaire reliability. After collecting data from the main questionnaire with 401 participants, the questionnaire's reliability was rechecked. Cronbach's alpha reliability test was conducted with all variables meeting the requisite threshold (0.6), except for self-responsibility (0.521). Despite this marginal difference, the variable was retained for analysis.

### 2.3. Participants

The sample comprised 401 participants from the general public who had visited nature sites at least once during the past year. In comparison to the Israeli Central Bureau of Statistics [54], the distribution of participants across these categories, as well as among education and income levels, mirrors the Israeli Central Bureau of Statistics [54], suggesting a representative cross-section of the Israeli populace (see Table 2); hence, no statistical correlation was conducted. However, Israeli Arabs were underrepresented in the sample, constituting only 18.5% compared to the 21.5% in the general population [54].

**Table 2.** Sociodemographic characteristics of the study participants.

| Variable | Category | Distribution (N = 401) | Proportion (%) |
|---|---|---|---|
| Gender | Male | 199 | 49.6 |
| | Female | 202 | 50.4 |
| Age | 18–24 | 58 | 14.5 |
| | 25–38 | 135 | 33.7 |
| | 39–52 | 91 | 22.7 |
| | 53–63 | 73 | 18.2 |
| | 64 and above | 44 | 11 |

**Table 2.** *Cont.*

| Variable | Category | Distribution (N = 401) | Proportion (%) |
|---|---|---|---|
| Religious affiliation | Jew | 327 | 81 |
| | Muslim | 50 | 12.5 |
| | Christian | 16 | 4 |
| | Druze | 8 | 2 |
| Self-definition of level of religiosity | Secular | 175 | 43.6 |
| | Traditional | 144 | 35.9 |
| | Religious | 51 | 12.7 |
| | Ultra-Orthodox | 30 | 7.5 |
| Education level | High school | 88 | 21.9 |
| | Tertiary or professional | 111 | 27.7 |
| | Academic BA | 132 | 32.9 |
| | Academic advanced degree | 68 | 17 |
| Income level | Far below average | 141 | 35.2 |
| | Below average | 83 | 20.7 |
| | Average | 101 | 25.2 |
| | Above average | 58 | 14.5 |
| | Far above average | 16 | 4 |

*2.4. Data Processing and Analysis*

Participants' responses were imported into SPSS V27 for subsequent data processing. Factor analysis was utilized to condense data into coherent factors. The relationship between self-declared littering frequency and sociodemographic attributes was explored through the chi-square test. Spearman correlations gauged associations between independent variables and the dependent variable (declared littering frequency). Pearson correlations examined relationships between littering attitudes, environmental attitudes, willingness to act, self-responsibility, and declared littering frequency.

Gender disparities in littering frequency, attitudes, willingness to act, and self-responsibility were scrutinized using independent sample t-tests, two-way ANOVA, and chi-square tests compared variable means. To discern the relative acceptability of littering-related behaviors, open-ended questions were deductively analyzed. Additionally, differences between self-littering and littering by others were tested using McNemar's test.

**3. Results**

This section is structured into four segments aligned with the research objectives. The initial segment will address perceptions and attitudes toward littering. The subsequent section will investigate potential associations between varied perceptions and demographic characteristics. The third section will examine rationales for littering behavior, along with motivations and effective strategies for promoting behavioral change. Lastly, the final part will analyze the public's preferred behaviors related to littering that are perceived as more acceptable.

*3.1. Linking Littering Behaviors in Open Spaces to Perception and Attitudes*

3.1.1. Attitudes, Willingness to Act, and Self-Responsibility

Participants exhibit a strong inclination to spend time in open spaces. Merely 9.5% (42 participants) out of the initial 443 participants did not engage in open-space activities throughout the past year. Consequently, the effective sample size is *n* = 401, as these 42 respondents could not complete the questionnaire. Despite lockdowns and COVID-19-related restrictions, a substantial majority (57.1%) of the final 401 participants visited open spaces multiple times during the year. Israeli participants attach great importance to the cleanliness of the open spaces they frequent; a notable 98.5% regard the cleanliness level as significant, with 13.5% deeming it important and 85% rating it as very important. Notably,

83% observed instances of littering by visitors in open spaces, and around 67% estimated that approximately half of open-space visitors engage in littering.

Respondents reported the cleanliness level of Israeli open spaces on a 1–5 Likert scale, ranging from very dirty (1) to very clean (5), with an average of slightly dirty (3) to clean (4) (M = 3.77, Std. = 0.864). Noteworthy cleanliness discrepancies emerged (F = 3.714, $p < 0.003$) among various types of open spaces: natural reserves with entrance fees and restricted hours (3.89 ± 0.765) and urban natural parks (3.98 ± 0.658) exhibited higher cleanliness than beaches (3.24 ± 0.903) and 24 h natural reserves (3.62 ± 0.945).

Anti-littering attitudes in open spaces exhibited a pronounced trend, with 80% favoring cleanliness (M = 4.49 ± 0.561). Conversely, the endorsement of environmental attitudes showed a notably lower average agreement (M = 3.67 ± 0.676). In essence, anti-littering attitudes surpassed environmental attitudes. Nevertheless, a robust positive correlation (Pearson correlation) existed among these attitudes (r = 0.537, $p < 0.0001$).

Participant agreement with statements reflecting a willingness to engage in cleanliness initiatives within open spaces (M = 3.45 ± 0.718) was less pronounced than their alignment with environmental attitudes, showcasing notable variability across responses. A similar pattern emerged with self-responsibility (M = 3.11 ± 0.856), which exhibited even lower agreement and greater variability. Consequently, a modest positive correlation (r = 0.146, $p < 0.003$) was observed between self-responsibility and anti-littering attitudes.

### 3.1.2. Self-Reporting on Littering Frequency and Habits

Most respondents identify as non-littering: A significant 80.3% claimed to have refrained from littering in any public space within recent years, with 16% admitting infrequent littering, 3% occasional littering, and 0.7% acknowledging regular public domain littering. Subsequently, participants were prompted to truthfully recount instances of littering in nature, as they were assured that occasional littering is commonplace. Of the participants, 128 individuals (32%) provided an account of such an occurrence. Remarkably, 28% of respondents confessed to never littering in public spaces, despite having engaged in such behavior (Figure 1). Furthermore, an equally notable statistic emerges: 48% of research participants, constituting every second individual within this representative Israeli sample, reported having littered in natural environments at least once over recent years (Figure 1).

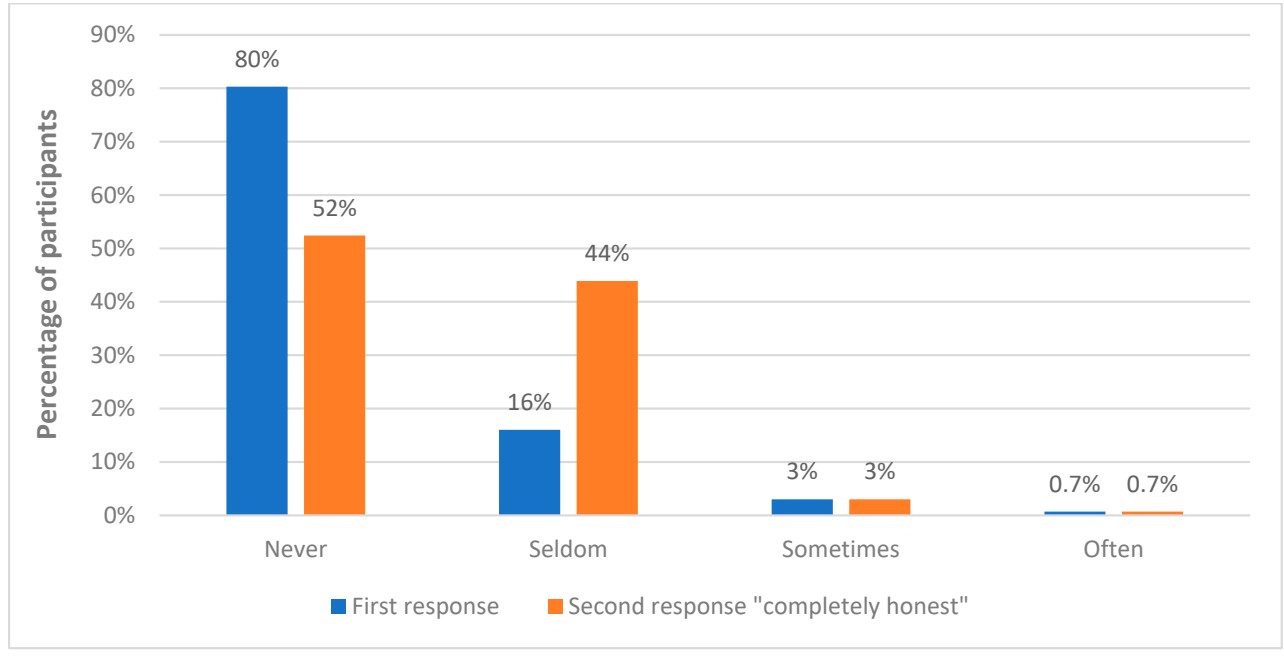

**Figure 1.** Reported littering frequency: comparison between the first and second rounds.

The emergence of three distinct clusters is observable in the analysis of nature littering perceptions (Figure 2). The initial cluster (52%) steadfastly maintains its assertion of never engaging in nature littering, even following the secondary inquiry. The subsequent cluster (28%) acknowledges past nature littering experiences despite their self-perception as non-litterers. The final cluster (20%) readily acknowledges infrequent, occasional, or frequent littering in both natural and other public settings.

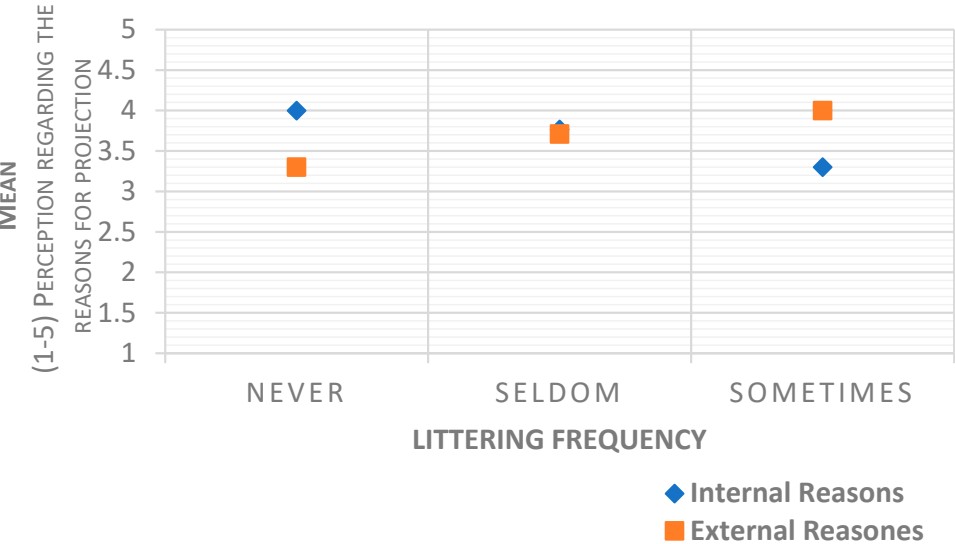

**Figure 2.** Differences in declared littering frequency as a function of the perception of the reasons for littering in nature. Values are mean ±std, as calculated from a 1–5 Likert scale.

### 3.2. Examining Sociodemographic Effecting Perceptions and Behavior toward Litter

Littering behavior frequency among the Israeli populace underwent scrutiny concerning sociodemographic attributes. A chi-square test and Spearman correlation were employed to analyze this association. A moderate negative correlation (r = −0.224, $p < 0.001$) emerged between age and littering frequency, indicating reduced littering with advancing age. A noteworthy relationship (X2 = 13.953, $p < 0.032$) materialized between self-identified religiosity and littering, with the ultra-Orthodox demographic displaying higher littering rates than the secular counterpart. Additionally, a weak negative correlation (r = −0.124, $p < 0.013$) materialized between education level and littering frequency, elucidating diminished littering rates with higher educational attainment. However, no significant correlations were discerned between gender, religious affiliation, income level, and self-reported littering frequency.

The variation in anti-littering attitudes, environmental attitudes, willingness to act, and self-responsibility of the Israeli public was examined with respect to the sociodemographic background data using a two-way ANOVA and a t-test for independent factors (Table 3).

Significant distinctions indicated by distinct lowercase letters near values within the same variable and sociodemographic traits signify statistical significance ($p < 0.05$). Notable divergences emerged in anti-littering attitudes by gender, age, and religiosity. Females (4.56 ± 0.5) exhibited stronger inclinations for cleanliness than males (4.4 ± 0.6), while younger individuals (ages 18–24) showed less alignment with anti-littering attitudes than older peers. Ultra-Orthodox respondents (4.13 ± 0.73) expressed reduced anti-littering alignment compared to the secular (4.52 ± 0.55), traditional (4.51 ± 0.54), and religious (4.52 ± 0.499) groups. No significant differences were observed in religious affiliation, education, or income.

**Table 3.** Sociodemographic factors affecting perceptions toward litter.

| Variable | Category | Anti-Littering Attitudes | | Environmental Attitudes | | Willingness to Act for Cleanliness | | Self-Responsibility for Cleaning | |
|---|---|---|---|---|---|---|---|---|---|
| | | Sig. | Mean ± std | Sig. | Mean ± std | Sig. | Mean ± std | Sig. | Mean ± std |
| Gender | Male | F = 5.00; p < 0.015 | b 4.4 ± 0.6 | NS | 3.63 ± 0.728 | NS | 3.44 ± 0.732 | NS | 3.15 ± 0.841 |
| | Female | | a 4.56 ± 0.5 | | 3.71 ± 0.621 | | 3.46 ± 0.705 | | 3.08 ± 0.872 |
| Age | 18–24 | F = 4.24; p < 0.002 | B 4.28 ± 0.64 | NS | b 3.23 ± 0.706 | F = 6.62; p < 0.001 | C 3.05 ± 0.69 | F = 3.208; p < 0.013 | b 3.12 ± 0.837 |
| | 25–38 | | a 4.60 ± 0.473 | | a 3.64 ± 0.629 | | b 3.51 ± 0.677 | | a 3.30 ± 0.802 |
| | 39–52 | | a 4.41 ± 0.63 | | a 3.76 ± 0.623 | | b 3.43 ± 0.719 | | b 3.07 ± 0.817 |
| | 53–63 | | a 4.51 ± 0.59 | | a 3.75 ± 0.706 | | b 3.49 ± 0.795 | | c 2.86 ± 0.899 |
| | 64 and above | | a 4.56 ± 0.437 | | a 3.95 ± 0.60 | | a 3.7 ± 0.594 | | b 3.02 ± 0.951 |
| Religious affiliation | Jew | NS | 4.46 ± 0.578 | NS | 3.62 ± 0.688 | F = 9.73; p < 0.001 | c 3.36 ± 0.696 | NS | 3.12 ± 0.88 |
| | Muslim | | 4.66 ± 0.434 | | 3.89 ± 0.605 | | a 3.90 ± 0.701 | | 3.12 ± 0.763 |
| | Christian | | 4.49 ± 0.529 | | 3.86 ± 0.244 | | b 3.55 ± 0.717 | | 2.83 ± 0.722 |
| | Druze | | 4.69 ± 0.561 | | 3.71 ± 0.676 | | b 3.84 ± 0.546 | | 3.50 ± 0.463 |
| Self-definition of level of religiosity | Secular | F = 4.38; p < 0.005 | a 4.52 ± 0.55 | NS | a 3.76 ± 0.606 | F = 8.98; p < 0.001 | a 3.52 ± 0.71 | NS | 3.07 ± 0.866 |
| | Traditional | | a 4.51 ± 0.54 | | a 3.70 ± 0.669 | | a 3.50 ± 0.732 | | 3.17 ± 0.85 |
| | Religious | | a 4.52 ± 0.499 | | a 3.58 ± 0.645 | | a 3.42 ± 0.598 | | 3.02 ± 0.798 |
| | Ultra-Orthodox | | b 4.13 ± 0.73 | | b 3.01 ± 0.804 | | b 2.81 ± 0.608 | | 3.34 ± 0.88 |
| Education level | High school | NS | 4.40 ± 0.63 | F = 6.92; p < 0.0001 | c 3.45 ± 0.717 | NS | C 3.22 ± 0.761 | NS | 3.12 ± 0.913 |
| | Tertiary or professional | | 4.44 ± 0.61 | | b 3.62 ± 0.723 | | b 3.42 ± 0.737 | | 3.10 ± 0.846 |
| | Academic BA | | 4.54 ± 0.485 | | a 3.72 ± 0.582 | | a 3.52 ± 0.668 | | 3.09 ± 0.866 |
| | Academic adv. degree | | 4.59 ± 0.506 | | a 3.92 ± 0.626 | | a 3.63 ± 0.656 | | 3.17 ± 0.792 |
| Income level | Far below average | NS | 4.49 ± 0.604 | NS | 3.61 ± 0.74 | NS | 3.42 ± 0.792 | NS | 3.14 ± 0.900 |
| | Below average | | 4.47 ± 0.521 | | 3.57 ± 0.647 | | 3.36 ± 0.704 | | 3.15 ± 0.841 |
| | Average | | 4.49 ± 0.552 | | 3.70 ± 0.668 | | 3.43 ± 0.712 | | 3.08 ± 0.872 |
| | Above average | | 4.53 ± 0.476 | | 3.80 ± 0.521 | | 3.56 ± 0.54 | | 3.12 ± 0.837 |
| | Far above average | | 4.49 ± 0.561 | | 3.99 ± 0.587 | | 3.80 ± 0.626 | | 3.30 ± 0.802 |

Similar trends surfaced in environmental attitudes regarding age and religiosity. Educationally, advanced degree holders showed the strongest environmental alignment (3.924 ± 0.626), followed by bachelor's (3.72 ± 0.582), tertiary/professional (3.616 ± 0.723), and high school (3.45 ± 0.717). No notable gender, religious affiliation, or income variations appeared.

Comparable patterns in the willingness to engage in cleanliness emerged across age, religiosity, and religiosity self-perception. "Young" participants had the lowest willingness (3.05 ± 0.69). Religious disparities influenced nature cleanliness willingness: Muslims (3.9 ± 0.701) exhibited higher inclination than Jews (3.36 ± 0.696). Ultra-Orthodox (3.01 ± 0.804) showed diminished engagement compared to other religiosity levels. A positive correlation linked education and willingness to act.

Self-responsibility was consistently lower than willingness and anti-littering attitudes. Only age impacted self-responsibility; the 25–38 group displayed elevated levels (3.30 ± 0.802).

### 3.3. Dissecting Rationales for Littering and Strategies for Mitigation

As per public opinion, the primary reason for littering in Israel is "intentional leisure-driven disposal" (M = 4.17, std = 1.165), whereas the least impactful factor is "absence of connection to the locale" (3.38 ± 1.274). Other factors contributing to littering were identified as follows, ranked in descending order according to mean scores: habitual behavior (4.09 ± 0.94), indolence (4.1 ± 0.97), negligence (4.17 ± 1.26), inadequate enforcement (4.03 ± 1.11), educational (4.07 ± 1.06) and informational deficits (3.61 ± 1.21), limited comprehension of ensuing harm (3.85 ± 1.15), disconnected sense of place (3.38 ± 1.26), and lack of concern for the environment (3.61 ± 1.18).

A two-way ANOVA was employed to assess variations in self-reported littering frequency concerning perceptions of reasons for littering, including both external, such as the availability of bins (F = 6.573, $p < 0.002$), and internal reasons (F = 10.412, $p < 0.000$), including a lack of concern, irresponsibility, and laziness. Those perceiving greater external influences for littering were primarily "sometimes" litterers (Mean = 4.00, std = 1.00), followed by infrequent litterers (3.71 ± 1.183) and non-litterers (3.30 ± 1.285) (Figure 2). Conversely, the pattern was inverted for internal reasons (Figure 3): individuals attributing higher significance to internal factors were more inclined to be non-litterers (4.0 ± 0.662), followed by occasional litterers (3.77 ± 0.681) and then those littering sometimes (3.30 ± 0.853).

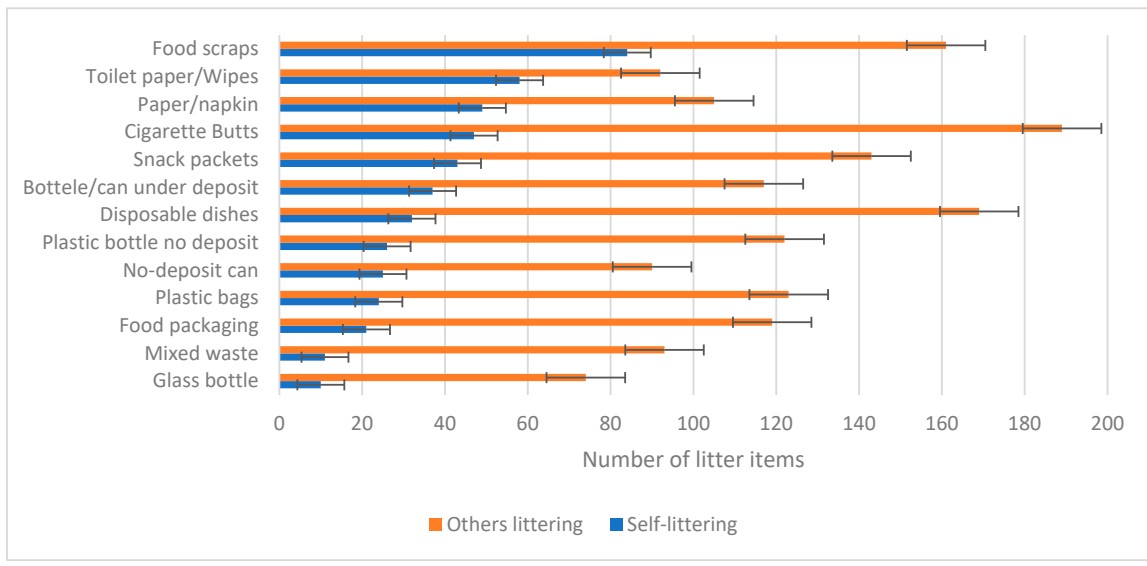

**Figure 3.** Comparison of litter items thrown away by the interviewee versus what was thrown away by others.

Regarding littering mitigation, participant perceptions highlight enforcement and fines (4.35 ± 0.98) as the most effective strategies, while payment upon nature site entry (2.47 ± 1.37) demonstrates limited efficacy. Notably, these two strategies significantly diverge from others—campaigns, media outreach (3.57 ± 1.183), early-age education (4.2 ± 0.99), infrastructure enhancement (4.2 ± 0.97), social norming (3.78 ± 1.126), and field-based initiatives (3.16 ± 1.22)—where noteworthy disparities are absent.

### 3.4. Unpacking Acceptable and Normative Littering Behaviors: Sub-Norm Categorization for Deeper Insight

While a majority (80%) of participants espoused anti-littering sentiments toward nature, a substantial 48% admitted to littering at least once, and 32% initially claimed non-littering but subsequently revealed past littering incidents. By scrutinizing their responses to open-ended inquiries detailing their littering conduct and motivations, insights can be gleaned into the littering behaviors participants deem more acceptable. Table 4 presents each littering behavior and its corresponding rationale that appeared in participants' open responses more than twice. The littering behaviors of individuals who engage in littering but do not identify as such (Group 2) and those who acknowledge occasional littering (Group 3) are outlined. It is notable that eight participants cited two distinct reasons, both enumerated separately. Notably, individuals perceiving themselves as non-litterers (Group 1) did not provide accounts of self-littering behaviors.

**Table 4.** Frequencies (number of mentions as a proportion of all mentions) of littering behaviors described by participants.

| Littering Behaviors and Justification | Group 2 (%) Litter in Nature but Do Not Perceive Themselves as Such | Group 3 (%) Litter in Nature Sometimes |
|---|---|---|
| Littering when bins are absent or remote | 37.5 | 69.4 |
| Collecting the waste in a bag and leaving it on the ground or a wall/tree | 15.8 | 2.0 |
| Organic degradable waste or organic waste (that feeds animals or biodegrades) | 10.0 | 8.2 |
| Personal factors: forgetfulness, laziness, lack of concentration, or being in a hurry | 6.7 | 10.2 |
| Litter blown with the wind or falling without being noticed | 6.7 | 4.1 |
| Leaving toilet paper and wipes behind after a "nature toilet" | 7.5 | 2.0 |
| The place is already dirty or other people are littering | 4.2 | 0.0 |
| Litter items that are small or difficult to collect such as melting ice cream or hot coals | 4.2 | 2.0 |
| Cigarette butts after smoking | 4.2 | 4.1 |
| Leaving litter beside a full bin | 3.3 | 0.0 |
| **Total** | **100** | **100** |

Littering behaviors manifest in diverse forms. Several are tied to external or environmental elements, like the presence of waste disposal infrastructure, such as bins. Others relate to litter type, including organic waste or toilet paper, while certain behaviors stem from personal factors like indolence, forgetfulness, or inattentiveness. Notably, some participants collect litter in plastic bags, either hanging them on trees or placing them beside full bins, reflecting an intent to avoid littering, despite their actions still contributing to litter accumulation. The participants tried justifying their littering behavior, and one of them wrote: "I had no choice, unfortunately there was no bin, and my vehicle was too far away to carry [the waste]". As perceived by the participants, their personal act of littering was construed as inevitable within their own framework of understanding; hence, it is deemed justifiable.

The survey asked participants about their personal littering and their observations of others' littering. McNemar's test revealed significant disparities between the amounts and

types of self-declared litter and observed litter by others ($p < 0.001$; Figure 3), potentially implying varying levels of acceptability for different litter items.

Cigarette butts dominate as the most frequently observed litter (Figure 3). Respondents predominantly self-reported littering food scraps, toilet paper, or wipes, whereas others were primarily observed littering cigarette butts, food scraps, and snack packets.

## 4. Discussion and Conclusions

This study aimed to analyze public perceptions in Israel concerning littering behavior within open spaces. This comprehensive investigation encompassed perceptions, willingness to act, and responsibilities, all contextualized by participants' demographic profiles. Additionally, this study delved into motives for littering and potential mitigation strategies and sought to identify littering behaviors deemed socially acceptable. The results highlight significant disparities between perceived and actual occurrences. Notably, a conspicuous example is the consensus favoring clean and well-maintained open spaces juxtaposed with the fact that 48% of participants reported recent instances of littering in public spaces [55–57]. Subsequently, the chapter will expound upon and derive conclusions in accordance with the research hypotheses.

**H1 Anti-littering attitudes and environmental viewpoints**: Our findings support the hypothesis that participants prioritize anti-littering attitudes over environmental perspectives. While numerous studies have explored littering behavior from the lens of pro-environmental conduct, a select few have unveiled connections between environmental perception and littering behavior. Individuals who identified as environmental advocates with favorable environmental attitudes demonstrated reduced littering and heightened maintenance of cleanliness [5,58]. Remarkably, direct evidence substantiating that the public widely aligns with anti-littering positions, nearly unanimously, over pro-environmental stances is lacking. Hence, it is plausible to hypothesize that the divergence observed in pro-environmental behavior studies between attitudes and actual behavior is more pronounced in the context of littering. Consequently, investigating this phenomenon warrants further scholarly exploration. Hence, efforts focused on altering attitudes will hold diminished pertinence in endeavors to modify littering behavior.

**H2 Anti-littering attitudes, willingness to act and self-responsibility**: This study's findings validate the hypothesis that the public exhibits stronger alignment with anti-littering perspectives compared to their willingness to act, and this divergence surpasses even the perception of individual accountability. This incongruence between willingness to act and attitudes aligns with earlier research [11,29]. Moreover, investigations addressing personal responsibility within environmental behavior demonstrate an inconsistent correlation between self-responsibility and the inclination to take action to translate intent into action [59]. Hence, it can be inferred that endeavors aimed at curbing habitual littering should prioritize enhancing the propensity for action.

**H3 Littering in the open spaces**: Our results concur with Hypothesis 3, elucidating that despite holding anti-littering attitudes, individuals continue to engage in littering within open spaces. Studies grounded in the planned behavior theory, elucidating littering behavior, underscore the potent association between attitudes and actions [60]. Our study affirms a robust agreement with anti-littering attitudes; however, such attitudes alone are insufficient to fully account for the observed behavior [61]. In an alternate study investigating the correlation between anti-littering attitudes and behavior in Israel, observations were recorded, extending beyond mere statements, revealing a reduced alignment between attitudes and actual littering behavior (Lev. et al., 2023; in process). Consequently, it is advisable to approach the assessment of attitudes cautiously when conducting research that seeks to drive action grounded in scientific foundations.

**H4 Attitude and behavior disparities across demographic facets**: While a diverse array of demographic groups acknowledged engaging in littering, our results validate that younger individuals are more prone to both admitting and participating in littering in comparison to their older counterparts [1,3,7,19]. Furthermore, in consonance with ear-

lier investigations, individuals with stronger religious affiliations manifest a heightened propensity for littering when contrasted with their secular peers [1,32,62]. Although our study identifies a relatively modest connection between littering and educational attainment, this association finds reinforcement from other inquiries [7,33]. Comprehending reference groups hold paramount significance for decision makers seeking emission reduction strategies. Tailoring educational approaches and advocacy based on group attributes can enhance effectiveness.

**H5 Conceptions of littering rationales and self-reported littering frequency**

This study's findings revealed a multifaceted array of motives attributed to public littering in open spaces. Interestingly, pinpointing a singular primary reason appears elusive. This phenomenon potentially elucidates the ambiguity surrounding the act of littering. Earlier literature similarly differentiated between environmental aspects (e.g., bin availability and cleanliness levels) and personal factors (e.g., lack of concern, irresponsibility, and laziness). The present study unveils a divergence in rationale perception between those admitting to littering, stratified by littering frequency. Those disposed toward littering evade self-condemnation, instead attributing responsibility to external factors. Conversely, individuals exhibiting hygiene mindfulness and self-assuredness are inclined to denigrate others, ascribing traits like indolence, ignorance, and obliviousness. Prior research [2,38] has indicated that when it comes to justifying self-littering behavior, internal personal factors, like laziness or a lack of attention, are less frequently employed, unlike external factors. However, these personal factors are often attributed to the "negative behavior" of others [29].

This connects to the last objective, which was to identify more "acceptable" and "normative" littering behaviors and categorize the general littering norm into sub-norms for deeper explanation. Evidently, individuals tend to rationalize littering in the absence of bins, and this is particularly prevalent among those who admit to littering in nature. This rationalization is often rooted in the perception that it is not their responsibility or that they lack alternative choices. Fewer instances of justification are noted when nature is already untidy, echoing findings that revealed a connection between littering and descriptive norms. Littered environments signal that there is no anti-littering in place; therefore, people are more inclined to litter [26] and are also less inclined to enforce transgressions of the anti-littering norm [23]. Social capital is also important, which encourages individuals to enforce norms [63,64].

**H6 Public preference for interventions dealing with littering**

This hypothesis was substantiated to a certain extent. This study's findings distinctly indicate that enforcement and fine imposition exert the most pronounced influence on littering behavior, marking a clear departure from other factors that exhibited negligible variations. Previous research has explored diverse interventions, including enforcement, revealing its potential effectiveness while acknowledging its potential to elicit resistance [65,66]. Furthermore, the efficacy of enforcement appears to diminish when it counters accepted norms [23]. Conclusively, a comprehensive approach encompassing diverse interventions and strategies is imperative to achieve successful littering reduction. While enforcement retains significance, relying solely on it proves ineffective.

Study Limitations: This study relies solely on self-reported statements, precluding real-time insight into actual behavior and thereby constraining the scope of the conclusions. Moreover, data collection occurred through an online questionnaire, potentially leading to the underrepresentation of ultra-Orthodox communities, which might be less inclined toward online engagement. Additionally, the questionnaire was administered in Hebrew, potentially introducing challenges for Arabic speakers during completion.

## 5. Recommendations

This study delineates evidence-based strategies pertinent to responsible authorities overseeing global and Israeli open-space management. Primarily, emphasizing the significance of implementing willingness to act perception tactics and nurturing environmental

accountability emerges as paramount. This shift advocates for a reduced reliance on knowledge for attitudinal transformation. To actualize this, targeted communication strategies catering to distinct demographics, like youth or the ultra-Orthodox, are recommended.

Decision makers should acknowledge that individuals who litter in open spaces rationalize their actions without self-inflicted negativity. Consequently, it is proposed to refine and subdivide established norms into sub-norms. For instance, when bins are scarce, carrying waste becomes imperative, as discarding it under such circumstances is regarded as unacceptable. Another recommendation involves enhancing the bag-based waste collection norm, accentuated by underscoring the importance of either carrying or disposing of it in designated enclosed bins for animal waste. Addressing waste types initially seen as more acceptable, like cigarette butts and organic litter, is pivotal. Paradoxically, discarding these contributes to pollution, endangering local fauna, and fostering an unsanitary environment that may inadvertently reinforce littering trends.

Site managers are urged to adopt a multifaceted approach instead of focusing solely on singular measures. This comprehensive approach, including enforcement, achieves heightened effectiveness when executed alongside well-maintained surroundings, suitable infrastructure, and a populace well-versed in established littering norms.

For improved research precision, further exploration is recommended, concentrating on delineating norms associated with littering. This can be accomplished by employing methodologies that blend observational data with exhaustive documentation of real-life behaviors.

**Author Contributions:** Conceptualization, N.L., M.N. and O.A.; Writing—original draft, N.L., M.N. and O.A. All authors have read and agreed to the published version of the manuscript.

**Funding:** Partial funding was received from the estate of Ernest Petrie.

**Institutional Review Board Statement:** The study was conducted in accordance with the Declaration of Helsinki, and approved by the Ethics Committee of the faculty of social sience, University of Haifa. (Protocol code 053/20 24.2.2020).

**Informed Consent Statement:** Informed consent was obtained from all subjects involved in the study.

**Data Availability Statement:** Data associated with this research is available upon request from the corresponding author and subject to any relevant legal or ethical restrictions.

**Conflicts of Interest:** The authors declare no conflict of interest.

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
