# Peer review of "Sometimes Littering Is Acceptable—Understanding and Addressing Littering Perceptions in Natural Settings"

_sustainability, doi:10.3390/su151813784_

Round 1
Reviewer 1 Report
The manuscript "Sometimes littering is acceptable – perceptions on littering in parks and open spaces" is interesting to readers. But before publication it requires changes to improve the manuscript and highlight the scientific contribution.
Abstract
The abstract must be rewritten. Because from the reading of the abstract, and will decide (when the topic is of interest) whether or not to continue reading the article depending on the clarity with which the author exposes his study in this first paragraph.
Authors should provide a summary of a study by answering the following questions:
- What was the overall purpose of the study?
- What was the specific objective of the study?
- Why was the study conducted?
- How was the study conducted?
- What did the study reveal?
Results
Change the title of table 3, it should be short, referring to the information presented.
Discussion
Is there any uncertainty of this study? If so, it is suggested to add in this Section
The discussion about results is weak and is not convincing,This section is poorly organized and the data have not discussed in a proper scientific manner.
Developing your discussion further by providing more in depth reflection.
Conclusion.
The conclusions are not strong enough. The analysis does not support it. The paper lacks any significant conclusions or recommendations
The conclusions presented are general and do not result from the conducted research.
Author Response
We wish to extend our gratitude for all reviewers’ diligent reviews. These comments and suggestions were incorporated into the manuscript and helped us to better convey the results and the conclusions. The feedback has proven to be immensely constructive, playing a pivotal role in refining the manuscript and consequently, yielding enhanced outcomes.
Below is a table summarizing the reviewers’ comments and the changes made. Please see the attachment for a more specific version of the table of changes.
Reviewer 1
|
|
|
|
|
1. |
Abstract - The abstract must be rewritten. Because from the reading of the abstract, and will decide (when the topic is of interest) whether or not to continue reading the article depending on the clarity with which the author exposes his study in this first paragraph. |
Abstract was rewritten taking under consideration the given advice Lines 5-30. |
|
2. |
Abstract – Authors should provide a summary of a study by answering the following questions: - What was the overall purpose of the study? - What was the specific objective of the study? - Why was the study conducted? - How was the study conducted? - What did the study reveal?
|
The rewritten abstract provides answers to these questions: What was the overall purpose of the study? Line 11-12 - What was the specific objective of the study? Line 13-16 - Why was the study conducted? Line 16-17 - How was the study conducted? Line 18-19 - What did the study reveal? Line 19-25
|
|
3. |
Results Change the title of table 3, it should be short, referring to the information presented.
|
Modified: Sociodemographic effecting perceptions towards litter Line 358 |
|
4. |
Discussion Is there any uncertainty of this study? If so, it is suggested to add in this Section
|
Added into the manuscript lines 556-561: Study Limitations: The study relies solely on self-reported statements, precluding real-time insight into actual behavior and thereby constraining the scope of conclusions. Moreover, data collection occurred through an online questionnaire, potentially leading to the underrepresentation of ultra-Orthodox communities, which might be less inclined towards online engagement. Additionally, the questionnaire was administered in Hebrew, potentially introducing challenges for Arabic speakers during completion.
|
|
5. |
Discussion The discussion about results is weak and is not convincing,This section is poorly organized and the data have not discussed in a proper scientific manner. |
This section was rewritten and edited according to objectives and hypothesis Lines 449-555. |
|
6. |
Discussion Developing your discussion further by providing more in depth reflection. |
This section was rewritten and edited according to objectives and hypothesis |
|
|
Conclusion. The conclusions are not strong enough. The analysis does not support it. The paper lacks any significant conclusions or recommendations
|
This section was rewritten regarding more specific conclusions Lines 563-586 |
|
|
Conclusion.The conclusions presented are general and do not result from the conducted research. |
This section was rewritten regarding more specific conclusions Lines 563-586 |
|
|
|
|
|
|
|
|

Reviewer 2 Report
The study explored the perception on littering in parks and open spaces and identify the gap between desired, postive attitudes and actual littering.
The title and abstract is clearly describe the study. The introduction part is well written with concise background of the study, people behavior regarding littering and its impact on health and environment. The methodology is sound and well executed. The results are presented in tanigble fashion and well discussed.
However, the authors didn't describe the sampling framework and how they reach 401 sample size?
Specific comments:
1. What is the main question addressed by the research? Comment: Identification of public's perception on littering in open spaces and parks in Israel.2. Do you consider the topic original or relevant in the field? Does it address a specific gap in the field? Comment: Yes, the topic is original and relevant to the field. It also addresses the gap in the field. Previous research focused on attitude and opinion towards littering on public spaces but didn't identify the perception of the public.
3. What does it add to the subject area compared with other published material? Comment: Yes
4. What specific improvements should the authors consider regarding the methodology? What further controls should be considered? Comment: The sampling procedure should be mentioned and which sampling method was used to reach the sample size of 401.
5. Are the conclusions consistent with the evidence and arguments presented and do they address the main question posed? Comment: Yes, the conclusion is in line with the data and argument presented in the results section and also addresses the main question.
6. Are the references appropriate? Comment: Yes
7. Please include any additional comments on the tables and figures. Comment: The tables and figures are presented in a decent way.
Minor spell check is required.
Author Response
We wish to extend our gratitude for all reviewers’ diligent reviews. These comments and suggestions were incorporated into the manuscript and helped us to better convey the results and the conclusions. The feedback has proven to be immensely constructive, playing a pivotal role in refining the manuscript and consequently, yielding enhanced outcomes.
Below is a table summarizing the reviewers’ comments and the changes made.
|
|
|
|
|
1. |
The study explored the perception on littering in parks and open spaces and identify the gap between desired, postive attitudes and actual littering. However, the authors didn't describe the sampling framework and how they reach 401 sample size? |
Sample design was added in chapter 2.1. lines 205-219 |
|
2. |
1. What is the main question addressed by the research? Comment: Identification of public's perception on littering in open spaces and parks in Israel. |
We added this sentence at the end of the introduction line 161-162. In addition, I made changes in the wording of the research goals and hypotheses |
|
3. |
2. Do you consider the topic original or relevant in the field? Does it address a specific gap in the field? Comment: Yes, the topic is original and relevant to the field. It also addresses the gap in the field. Previous research focused on attitude and opinion towards littering on public spaces but didn't identify the perception of the public. |
Thank you for the syllable. We added this reference in line 169 |
|
4. |
3. What does it add to the subject area compared with other published material? Comment: Yes |
The following text was added – צייני איפהThe study highlights and deepens the public's perceptions of the subject. He renews the comparison made between the perceptions related to littering in front of the environmental perceptions, the holocaust between the positions, the desire to act and the personal responsibility. It also sharpens behaviors and norms that are considered acceptable for litterers in the wild. All the above is mentioned in the rewritten introduction |
|
5. |
What specific improvements should the authors consider regarding the methodology? What further controls should be considered? Comment: The sampling procedure should be mentioned and which sampling method was used to reach the sample size of 401. |
The sample design was added in chapter 2.1. Lines 205-219 |
|
|
5. Are the conclusions consistent with the evidence and arguments presented and do they address the main question posed? Comment: Yes, the conclusion is in line with the data and argument presented in the results section and also addresses the main question. |
The objectives were rewritten and hypotheses were added to the text. The results section was reorganized by the four objectives. |
|
|
6. Are the references appropriate? Comment: Yes |
|
|
|
7. Please include any additional comments on the tables and figures. Comment: The tables and figures are presented in a decent way. |
|
|
|
Minor spell check is required |
All text was added, rephrased and spell-checked. |
|
|
|
|

Reviewer 3 Report
The manuscript is well-written, methodologically sound, and presents interesting results of considerable relevance. Furthermore, it aligns well with the scope of Sustainability.
However, there are some issues that need to be addressed before the manuscript can be considered for publication.
*Major
Literature.
When discussing norms it might be worth adding more recent research, which extends the more received research that has been cited (e.g. Cialdini et al. 1990). Littered environments, in contrast to clean ones, signal that there is no anti-littering in place Therefore, people are more inclined to litter (Cialdini 1990), and also less inclined to enforce transgressions of the anti-littering norm (Berger & Hevenstone). Also important is social capital, which encourages individuals to enforce norms (Botetzagias et al. 2020, Berger 2021).
Botetzagias, I. et al. (2020) ‘Exercising social control in PAYT (Pay-As-You-Throw) violations: The role of subjective evaluations and social capital’, Waste Management, 105, 347–354. Available at: https://doi.org/10.1016/j.wasman.2020.02.020.
Berger, J., Hevenstone, D. (2016) ‘Norm enforcement in the city revisited: An international field experiment of altruistic punishment, norm maintenance, and broken windows’, Rationality and Society, 28(3), 299–319. Available at: https://doi.org/10.1177/1043463116634035.
Berger, J., & Hevenstone, D. (2016). Norm enforcement in the city revisited: An international field experiment of altruistic punishment, norm maintenance, and broken windows. Rationality and Society, 28(3), 299-319.
Cialdini, R.B., Reno, R.R., Kallgren, C.A. (1990) ‘Cialdini et al. (1990) - A Focus Theory of Normative Conduct’, Journal of Personality and Social Psychology, 58(6), 1002-1012.
Methods and results.
When reproting Cronbach’s Alpha, please refer to a correlation matrix (e.g. in an appendix).
It is stated in the manuscript that, regarding index building, “adjustments were made as required”. Please explain which kinds of adjustments have been wade for what reason. All results must be reproducible and this requires a clear description of your plan of analysis.
Data processing and analysis. Primarily, t-tests are used. Would it not be preferable to conduct multivariate regressions or ANOVA? Typically, multivariate methods are preferable for non-experimental data to obtain the net effects of potential explanatory factors. If the authors have good reasons to only use bivariate analyses, they should rationalize their strategy carefully.
Figure 3 and 4: Please report 95%-confidence intervals along with the bars.
*Minor
There’s a typo in subheading 1.4 . Objectives (to many spaces after point)
There’s a point missing at the end of a sentence on p, 17, l 394: This study emphasizes the gap between visitors’ desire for clean open spaces and 394 parks and the actual cleanliness level caused by littering behavior (Björnsdóttir, 2018; Ca- 395 pacci et al., 2015; Göktuğ, 2021)
Good, some typos.
Author Response
We wish to extend our gratitude for all reviewers’ diligent reviews. These comments and suggestions were incorporated into the manuscript and helped us to better convey the results and the conclusions. The feedback has proven to be immensely constructive, playing a pivotal role in refining the manuscript and consequently, yielding enhanced outcomes.
Below is a table summarizing the reviewers’ comments and the changes made
|
|
The manuscript is well-written, methodologically sound, and presents interesting results of considerable relevance. Furthermore, it aligns well with the scope of Sustainability. |
Thank you. |
|
1. |
Literature. When discussing norms, it might be worth adding more recent research, which extends the more received research that has been cited (e.g. Cialdini et al. 1990). Littered environments, in contrast to clean ones, signal that there is no anti-littering in place Therefore, people are more inclined to litter (Cialdini 1990), and also less inclined to enforce transgressions of the anti-littering norm (Berger & Hevenstone). Also important is social capital, which encourages individuals to enforce norms (Botetzagias et al. 2020, Berger 2021).
|
Reference which reinforces Cialdini classic research were add in line 19 and line 499
The recommended references were added to literature: Botetzagias, I. et al. (2020) Berger, J., Hevenstone, D. (2016) |
|
2. |
Methods and results. When reproting Cronbach’s Alpha, please refer to a correlation matrix (e.g. in an appendix).
|
All the details about the correlation and the data appear in Table 2 which describes the variables of the study |
|
3. |
It is stated in the manuscript that, regarding index building, “adjustments were made as required”. Please explain which kinds of adjustments have been made for what reason. All results must be reproducible and this requires a clear description of your plan of analysis. |
Added in line 257-258: leading to adjustments that aligns with the language and terminology commonly employed in Israel |
|
|
Data processing and analysis. Primarily, t-tests are used. Would it not be preferable to conduct multivariate regressions or ANOVA? Typically, multivariate methods are preferable for non-experimental data to obtain the net effects of potential explanatory factors. If the authors have good reasons to only use bivariate analyses, they should rationalize their strategy carefully. |
The identification of variations in perceptions based on demographic data, along with distinctions in the understanding of dumping rationales corresponding to waste disposal statements, was accomplished using ANOVA, as outlined in lines 279, 357, and 392. |
|
|
Figure 3 and 4: Please report 95%-confidence intervals along with the bars.
|
Figure 3 confidence interval was added. Figure 4 was deleted from the manuscripts. |
|
|
There’s a typo in subheading 1.4 . Objectives (to many spaces after point) |
All objectives were rephrased |
|
|
There’s a point missing at the end of a sentence on p, 17, l 394: This study emphasizes the gap between visitors’ desire for clean open spaces and 394 parks and the actual cleanliness level caused by littering behavior (Björnsdóttir, 2018; Ca- 395 pacci et al., 2015; Göktuğ, 2021) |
Thank you for noticing. Modified. |

Round 2
Reviewer 3 Report
The authors have made significant improvements to the manuscript, and it is now ready for publication.